



# Coastal ocean response during the unprecedented marine heat waves in the western Mediterranean in 2022

Mélanie Juza[1](mjuza@socib.es), Marta de Alfonso[2], Ángels Fernandez-Mora[1]

[1] *Balearic Islands Coastal Observing and Forecasting System (SOCIB), Palma, Spain*
[2] *Puertos del Estado, Madrid, Spain*

## **Abstract**

The western Mediterranean Sea suffered unprecedented marines heat waves (MHWs) in 2022. This study focuses on the response of coastal ocean, which is highly vulnerable to global warming and extreme events that threaten the biodiversity, as well as goods and services that humans rely on. Using remote sensing and in situ observations, strong spatio-temporal variations of MHWs characteristics are observed in the coastal ocean over the last decade 2013-2022. In 2022, shallow-water moorings in the western Mediterranean Sea detected between 23 and 131 days of MHWs. While the highest MHW mean and maximum intensities were detected at surface in the French waters, the highest duration was observed near-shore at 17 m depth in the Balearic Islands. As thermal stress indicators for marine ecosystems, the cumulative intensity and total days were found the highest at surface at Tarragona, and MHW temperatures warmer than 28ºC were observed up to 58 days at Palma. Differences between satellite products and moorings observations, as well as between daily and hourly in situ measurements are also highlighted inviting us to continue the efforts in deploying and maintaining multi-platform observing systems in both open and coastal ocean waters to better address the coastal adaptation and mitigation in the context of climate change.

## **Introduction**

The Mediterranean Sea is one of the most vulnerable regions to climate change and responds rapidly to global warming with strong spatial variations (Giorgi, 2006; Lionello and Scarascia, 2018; Juza and Tintoré, 2021a; Juza et al., 2022). In 2022, the western Mediterranean Sea (WMed) suffered extreme ocean temperatures and several marine heat waves (MHWs) in a row from May to December 2022 (Juza and Tintoré, 2020, 2021b). These MHWs were exceptional for their early occurrence, intensity, duration and spatial extent. In the Balearic Islands region, the warmest spatially-averaged satellite sea surface temperature (SST) ever registered since 1982 was observed on the 13th of August 2022 with a value of 29.2 ºC, corresponding to an anomaly of 3.3 ºC with respect to the period 1982-2015, exceeding the previous regional record in summer 2003 (Juza and Tintoré, 2020, 2021b). Warmer



temperatures and anomalies can be found locally than regionally due to their strong spatial
variations (Juza and Tintoré, 2021a). In summer 2022, ocean temperatures reaching more than
32ºC were observed in the Mallorca Channel[1], while SST anomalies exceeded 5ºC in French
waters, reaching historical records ever registered since 1982 (Guinaldo et al., 2023).

37         The Mediterranean Sea is the largest semi-enclosed sea, with 46.000 km of coastline

and many islands, being also considered a hot-spot of biodiversity with many endemic species
(Coll et al., 2010). Its coastal zone provides goods and services that humans rely on (Smith et
al., 2021; UNEP/MAP and Plan Bleu, 2020) but it concentrates and accumulates human
pressures (e.g. contamination, population in cities, overfishing, coastline artificialization,
marine traffic, offshore industry and tourism) (UNEP/MAP and Plan Bleu, 2020). In addition,
the coastal areas and ecosystems are highly vulnerable to global warming and extreme
temperature events that threaten the biodiversity in the Mediterranean Sea (Cerrano et al., 2000;
Garrabou et al., 2009, 2019, 2022; Bensoussan et al., 2019; Verdura et al., 2019). Recently,
Garrabou et al. (2022) have shown that MHWs drive recurrent mass mortalities of marine
organisms in the Mediterranean Sea. These mass mortality events affected thousands of
kilometres of coastline from the surface to 45m, across a range of marine habitats and taxa.
Also, *Posidonia Oceanica*, which is the dominant seagrass in the Mediterranean Sea living
between surface and 40m depth, is very sensitive to high temperatures above 27ºC, particularly
in its early stage of development (Guerrero-Meseguer et al., 2017). Verdura et al. (2021) also
highlighted during the 2015 event high mortalities of habitat-forming seaweeds at temperatures
of 28ºC with most severe implications for early life stage and fertility. In 2017, concomitant
with the thermal context, the large-scale and long-lasting mucilaginous benthic algal bloom
was observed along the coasts of the northern Catalan Sea affecting benthic coastal habitats
(Bensoussan et al., 2017).

57         The climate signal manifests differently from coastal areas to the open ocean and in the

different sub-regions due to the variety and complexity of coastal ocean processes (Juza et al.,
2022). Satellite products and *in situ* measurements are complementary ocean data sources.
There is a benefit of using *in situ* data as a complement of satellite products since they provide
a more accurate representation of the thermal characteristics in the near-shore environment
(Schlegel et al., 2017a). Satellite data are not always accurate close to the land and have a lower
temporal resolution. In this study, the coastal ocean response to the unprecedented MHWs that

---

[1] SOCIB news [10-08-2022]: https://www.socib.es/index.php?seccion=detalle_noticia&id_noticia=535, last access: 19 June 2023.





occurred in the WMed in 2022 is analysed using daily data from satellite observations and
coastal mooring measurements. Then, the events detected by moorings in 2022 are compared
to those observed over the last decade since 2013. In addition, since MHW events are addressed
in coastal areas where ecosystems are highly present and sensitive, the range of temperatures
reached during these events is also studied, in particular MHW temperatures exceeding 28ºC,
when strongly altering marine habitat and accelerating species mortality. Finally, these extreme
temperature ranges are investigated through the analyses of daily and hourly data highlighting
differences in thermal stress estimations.
**Datasets and methodology**
Datasets
Daily reprocessed (REP) and near real-time (NRT) satellite products in the Mediterranean Sea
distributed by the Copernicus Marine Service[2] are used (products ref. no. 1 and 2, Table 1).
These products provide optimally interpolated estimates of SST into regular horizontal grids
of 1/20º and 1/16º spatial resolutions, respectively, covering the period 1982-2022 (Pisano et
al., 2016; Buongiorno Nardelli et al., 2013).
Hourly temperature timeseries from moorings in the WMed were uploaded from the
Copernicus Marine In Situ data portal[3] (product ref. no. 3, Table 1) and the Balearic Islands
Coastal Observing and Forecasting System (SOCIB) data catalogue[4] (products ref. no. 4 and
5, Table 1). Fixed stations with data covering the period 2013-2022 with limited temporal gaps
have been selected. In addition, focusing the study on the coastal response to extreme
temperature events, deep water stations (off the continental shelf) have been excluded. A total
of 10 coastal moorings located at depths shallower than 200 m are used in this study (Table 2,
Figure 1). Finally, all moorings data were post-processed removing spikes and erroneous data.
Methodology
The commonly used methodology for MHW identification and characterization from Hobday
et al. (2016) is applied. MHWs correspond to daily SSTs exceeding the daily 90th percentile of
the local SST distribution over a long-term reference period during at least five consecutive
days. In addition, two successive MHW events with 2-day or less time break are considered as
a continuous event. This also allows discarding the unrealistic jumps in SST time series due to
sparse erroneous daily interpolated data in the NRT satellite product or in temperature time

---

[2] Copernicus Marine Service: https://marine.copernicus.eu/, last access: 19 June 2023.
[3] CMEMS In Situ TAC:http://www.marineinsitu.eu/, last access: 19 June 2023.
[4] SOCIB thredds catalog: https://thredds.socib.es/thredds/catalog.html, last access: 19 June 2023.



series from *in situ* measurements. Finally, the daily climatological mean and threshold time
series are smoothed using a 30-day moving window to extract useful climatology from
inherently variable data.
First, daily SST from satellites are used to compute climatology over the period 1982-2015 and
to detect MHWs from 1982 to 2022, providing valuable information about the 2022 thermal
situation over the whole Mediterranean. The chosen reference period starts as early as possible,
covers at least a 30-year period as recommended (Hobday et al., 2016) and is aligned with the
methodology applied in recent publications in the Mediterranean Sea (Juza and Tintoré, 2021;
Juza et al., 2022). Then, the computation and detection are applied to the daily mean
temperature timeseries from mooring and the nearest satellite point when *in situ* data are
available, both over the commonly available period 2013-2022 for their direct comparison.
Although the *in situ* time series are shorter than the recommended 30-year minimum for the
calculation of climatology and characterization of MHWs, the calculation of MHWs using their
own climatology allows quantifying the amount they differ from their localities (Schlegel et
al., 2017b; Juza et al., 2022).
MHW indices are then calculated to characterize the 2022 MHW event and to estimate changes
over the last decade. For each year, the MHW mean and maximum intensities above the mean
climatology, mean duration and number of discrete events are computed. MHW cumulative
intensity and total days are also provided as interesting indicators for ecosystem stressor,
although they are an aggregation of MHW intensity and duration, and of duration and
frequency, respectively. Finally, ocean temperatures exceeding 28ºC are also identified during
the detected MHW events. The combination of abnormal conditions (MHW) and stressful
threshold (temperature ranges) allows identifying high thermal stress situations that strongly
impact marine ecosystems. In this respect, these extreme temperatures are also investigated
through the use and analysis of hourly data as observed by the moorings.
**MHWs in the Mediterranean Sea**
MHWs are firstly detected using satellite SST with respect to the reference period 1982-2015.
MHW characteristics are quantitatively sensitive to the baseline period but remain qualitatively
consistent (Dayan et al., 2023). All MHW characteristics are substantially increasing in the
Mediterranean Sea over the last decades, as studied over 1982-2020 (Juza et al., 2022), 1987-
2019 (Dayan et al., 2023) and 1982-2021 (Pastor and Khodayar, 2023). Over the recent period
1982-2022, the local trend estimates with 95% confidence for the MHW characteristics have





reached maximum values of MHW mean and maximum intensities, mean duration, frequency
and total days of 0.18 and 0.65°C/decade, 12.4 days/decade, 2.4 events/decade and 42.2
days/decade, respectively (Juza and Tintoré, 2021b, Vargas-Yáñez et al., 2023). In 2022,
annual mean and maximum intensities, mean duration, frequency and total days in the whole
Mediterranean oscillate locally over 0.95-3.10 and 1.24-6.47°C, 5-235 days, 1-15 events and
5-291 days, respectively (Figure 2A for MHW total days). In 2022, there are strong differences
in MHW characteristics between the western and eastern sub-basins. In the WMed,
unprecedented MHWs occurred in 2022 which was the year with the highest annual total days
of MHWs over the period 1982-2022 reaching up to 291 days locally along the Spanish coast
in the Balearic Sea (Figure 2A). Spatially integrated in the WMed, annual MHW characteristics
reached records ever registered since 1982 during the year 2022 (Figure 2B for MHW total
days). In particular, mean and maximum intensities, mean duration and total days reached 2.25
and 4.36°C, 36.6 and 180 days, respectively.
**Coastal MHWs in 2022**
MHWs are then detected from daily temperature from mooring and satellite with respect to the
reference period 2013-2022, which is the longest common period available in the moorings of
study. The use of shorter time series for climatology induces errors in MHW detection and
characterization, in particular due to ocean warming trend (Juza et al., 2022; Izquierdo et al.,
2022). More precisely, MHW characteristics detected by satellites at the nearest point from
moorings differ according to the reference period used (not shown). Since the SST
climatologies have higher values over 2013-2022 than 1982-2015, fewer MHW events are
detected using the 2013-2022 reference period. More specifically, annual MHW total days,
maximum and cumulative intensities are underestimated by at least 21, 5 and 29%,
respectively, according to the year and mooring location over 2013-2022, and up to 100% some
years when MHWs are not detected with the recent and short reference period for climatology
(Table 3).
Results from moorings
In 2022, all moorings of the WMed detected MHWs over the last decade (Figure 3), although
MHWs were computed using the reference period 2013-2022. As mentioned above, the use of
recent baseline periods underestimates these extreme events (Table 3) due to ocean warming.
Different responses are highlighted between the moorings (Figure 3, Table 4), not only because
of the different depths of sensor installation but also because of their geographical location.





Indeed, results from satellite data at the nearest point also indicate the strong spatial variability.
In 2022, the highest mean and maximum intensities of MHWs detected by moorings are found
along the French coast (Sète and Leucate) and the southern Spanish coast (Malaga) up to 3.67
and 5.17ºC, respectively. The highest mean duration is detected in the near-shore moorings at
Cala Millor (40 days) and Son Bou (31 days) installed at 17 m depth, as well as in the coastal
Balearic Sea (Tarragona, Dragonera and Palma) where the highest total days is observed with
values up to 131 days at Tarragona in 2022. Such responses have led to highest cumulative
intensity and possibly associated thermal stress on ecosystems in the moorings at Palma,
Dragonera, Tarragona, Sète and Leucate. Finally, MHW days with temperature exceeding 28ºC
are found in the Balearic Sea, from Barcelona to Cala Millor and Son Bou, with the highest
numbers at Tarragona (47), Dragonera (53) and Palma (58). In addition, these highly stressful
thermal situations with temperatures higher than 28ºC occurred several times during the
summer 2022 with long periods of consecutive days (up to 33 days at Palma). Moorings located
along the French coast (Leucate and Sète) and in the Alboran Sea (Malaga and Melilla) did not
face daily temperatures warmer than 28ºC.
Differences with satellite
Differences between moorings and satellites are found in all locations although the satellite
points are very close to corresponding moorings (Table 4). In 2022, along the French coast,
moorings observed higher MHW mean intensity at Sète and Leucate (by 0.39 and 0.23ºC,
respectively) and higher MHW maximum intensity at Leucate (by 1.47ºC) than satellites. On
the contrary, satellites detected higher MHW mean and maximum intensity at Barcelona than
moorings, with differences around 0.5 and 1.07ºC, respectively. Strong differences in MHW
maximum intensities are also found at Melilla, Palma and Son Bou (by 1.13, 0.53 and 0.52ºC
respectively). The MHW mean duration is found longer in moorings than satellites particularly
at Cala Millor, Son Bou and Tarragona (by 15, 10.3 and 7.9 days, respectively) while it is
particularly longer in satellites than in moorings at Dragonera and Palma (by 8.3 and 13.4 days,
respectively). The MHW total days and cumulative intensity in 2022 are higher in moorings at
Sète and Tarragona than in satellites at the nearest point while they are found higher in satellites
at Leucate, Barcelona, Balearic Islands stations (particularly at Cala Millor and Son Bou) and
Melilla. Finally, where MHW days with temperatures warmer than 28ºC are found (from
Barcelona to Son Bou), the number of days is higher in satellites than in mooring, except at
Tarragona.





Differences between MHWs detected by satellites and moorings may be explained by the depth
of measurements. While satellites provide SST, the selected moorings collected temperatures
at surface or subsurface (from 0.4 to 17 m depths, Table 2). However, even for moorings with
sensors installed near the surface (up to 0.5 m), strong differences with satellites are pointed
out as found at Sète, Leucate and Barcelona for MHW mean and maximum intensities (up to
0.5 and 1.47 ℃, respectively), and at Tarragona for MHW mean duration (13.4 days). Also,
importantly, results at Cala Millor and Son Bou strongly differ between satellites at the surface
and moorings in subsurface (particularly in MHW total days and days with temperature warmer
than 28ºC), as well as, between satellite locations and between moorings highlighting the
coastal ocean response differ from surface to subsurface and from one location to another at
both surface and subsurface even in the same sub-region (on each side of the Menorca Channel
in the Balearic Islands).

## Coastal MHWs from 2013 to 2022

MHWs observed by the moorings are now analysed from 2013 to 2022 and the events in 2022
are compared with those over the last decade (Figure 4). All years over 2013-2022 suffered
MHWs in several locations of the coastal WMed. In 2020 and 2022, all moorings detected
MHWs. While 2020 events mostly happened in winter, 2022 MHWs mainly occurred in
summer reaching high ocean temperatures.
Time series of annual MHW characteristics from moorings show strong spatio-temporal
variability. Variations in MHW mean and maximum intensities are highlighted between years
while the increase in MHW frequency and duration in recent years leads to a clear increase in
MHW total days and cumulative intensity. In recent years, MHWs did not only occur during
their usual season over a longer period but also extended over more seasons. When one season
was concerned in 2013, MHW occurrences covered three seasons in 2022 (not shown).
The analysis over the period 2013-2022 highlights that many thermal records were reached in
2022. MHW total days reached the highest number in 2022 for the stations at Leucate,
Barcelona, Tarragona, Dragonera, Palma, Cala Millor, the second highest at Sète, Son Bou,
Melilla and the fourth highest at Malaga. The MHW cumulative intensity in 2022 is the
warmest observed since 2013 for the stations at Leucate, Barcelona, Tarragona, Dragonera,
Palma, Cala Millor, Melilla, the second warmest at Sète and Son Bou, and the third warmest at
Malaga. In addition, in 2022, the number of MHW days with temperatures exceeding 28ºC is
the highest and can be considered as the unique year until now for the moorings at Barcelona,



Tarragona, Dragonera, Palma, Cala Millor, Son Bou, although Palma and Tarragona also
experienced 7 and 5 days, respectively, with such warm temperatures in 2015.
**Discussion**
Hourly measurements from moorings were averaged on a daily basis to be compared with the
daily satellite products. The associated standard deviations over 2013-2022 oscillate between
0.23 and 0.39 ºC depending on the stations. In this section, the temporal resolution impact on
the estimation of thermal stress during MHW events is analysed, in particular when high
temperatures of 28ºC or more are reached. As highlighted above, the MHW events concerned
are those in 2022 at the moorings from Barcelona to Son Bou.
Due to the diurnal cycle, maxima of MHW temperatures are found in the hourly datasets
(Figure 5). While the maxima from the daily datasets vary between 28.37ºC (Barcelona) and
29.95ºC (Palma), in the hourly datasets they oscillated between 28.96ºC (Cala Millor) and
31.36ºC (Dragonera), this latter being the record ever registered by the Spanish mooring
network from Puertos del Estado. The difference between the daily and hourly data maxima is
the highest at Dragonera (1.52ºC) and the smallest at Palma (0.05ºC). The distribution of the
temperatures higher than 28ºC is schematically represented by the median, as well as the 5 and
95[th] percentiles whose difference allows estimating the width (Figure 5). This latter is larger in
the hourly than daily datasets due to the diurnal cycle. Comparing the moorings between
themselves, the width is larger in both daily and hourly datasets at Dragonera (1.34 and 1.56ºC,
respectively), Palma (1.33 and 1.42ºC, respectively) and Tarragona (1.07 and 1.30ºC,
respectively) where warmer temperatures were reached.
At Palma, the daily and hourly data provide similar results on the maxima reached and
distribution characteristics of extreme ocean temperatures in summer. At the moorings located
further off the coast of peninsula (Barcelona, Tarragona and Dragonera), the temporal
resolution of *in situ* data clearly impacts the extreme temperature observations. Such findings
are also highlighted in the two near-shore stations although their sensors are located at 17m
depth.
**Conclusions**
Society is facing unprecedented challenges arising from climate change impacts. Among them,
marine heat waves (MHWs) are becoming more frequent, longer and more intense worldwide
(Frölicher et al., 2018, Oliver et al., 2018) and particularly in the Mediterranean Sea (Juza et



al., 2022; Dayan et al., 2023; Pastor and Khodayar, 2023). Such physical changes have major
ecological impacts with socio-economic implications and compromising carbon storage,
particularly in coastal ocean waters (Smith et al., 2021, 2023). Although MHWs are mainly
induced by large-scale anomalous atmospheric conditions in the Mediterranean Sea (Holbrook
et al., 2019; Guinaldo et al., 2023; Hamdeno and Alvera-Azcarate, 2023), the ocean response
strongly differs from the open ocean to near-shore areas, and from one coastal location to
another.
In this study, MHWs in the coastal and shallow waters of the western Mediterranean Sea
(WMed) have been investigated during the year 2022 and the period 2013-2022. Satellite and
moorings observed MHWs along the coast of the WMed whose characteristics strongly vary
in time and space. Coastal MHWs were observed almost every year over the last decade, and
they were exceptional in 2022 in intensity, duration and geographical extension. In 2022,
although the coastal MHW events have a strong spatial variation, all moorings - from northern
to southern WMed, from surface to subsurface - observed MHWs registering records in
intensity (in French waters), duration (in subsurface in the Balearic Islands), total days,
cumulative intensity (at Tarragona), and number of days with temperature warmer than 28ºC
(at Dragonera and Palma).
Although the satellite products have the great benefit to monitor all the ocean surface,
differences with the moorings have been detected in the characterization of MHWs in coastal
areas and shallow waters. Compared with mooring measurements at surface (between 0 and
3m depth) in 2022, satellites underestimate MHW intensities in French waters and MHW
duration at Tarragona while they overestimate MHW intensities at Barcelona, Palma and
Melilla, as well as MHW duration at Dragonera and Palma. The thermal stress on the physical
and biological oceans is also minimized with the use of daily data compared to hourly
measurements. Finally, the ocean response to extreme warm events strongly differs from one
location to another even in the same region, from surface to subsurface even in very shallow
waters. Such findings assert the importance of multi-platform, multi-sensor and sustainable
ocean observing systems from open to coastal and near-shore waters and from surface to
subsurface to continue the investigation concerning MHWs and impact assessment.

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



**Tables**

| Product ref. no. | Product ID & type | Data access | Documentation |
|---|---|---|---|
| 1 | SST_MED_SST_L4_NRT_OBSER VATIONS_010_004 (1982-2021); Satellite observations | EU Copernicus Marine Service Product, 2022a | Quality Information Document (QUID): Pisano al., (2022a)<br><br>Product User Manual (PUM): Pisano et al., (2022b) |
| 2 | SST_MED_SST_L4_REP_OBSER VATIONS_010_021 (2022); Satellite observations | EU Copernicus Marine Service Product, 2022b | Quality Information Document (QUID): Pisano al., (2022c)<br><br>Product User Manual (PUM): Pisano et al., (2022d) |
| 3 | INSITU_MED_PHYBGCWAV_DI SCRETE_MYNRT_013_035 (2013-2022); In Situ observations | EU Copernicus Marine Service Product, 2022c | Quality Information Document (QUID): Wehde et al., (2022)<br><br>Product User Manual (PUM): In Situ TAC partners (2022) |
| 4 | Buoy Bahia de Palma_Physico-chemical parameters of sea water data (2013-2022); In situ observations | Balearic Islands Coastal Observing and Forecasting System (SOCIB) product, 2022 | Tintoré, J. (2022) |
| 5 | Two nortek AWACs in near-shore Balearic Islands; In situ observations (extended until 2022) | Balearic Islands Coastal Observing and Forecasting System (SOCIB) data, 2022 | Fernández-Mora et al., (2021) |

435            *Table 1: Product Table describing data products used in this study.*

| Mooring | Nº | Location Mooring | Location Satellite | Distance (km) | Sensor depth (m) | Bathymetry (m) |
|---|---|---|---|---|---|---|
| Sète | 1 | 43.37ºN-3.78ºE | 43.35ºN-3.77ºE | 1.8 (SSW) | 0.0, 0.4 (since 2019-04-16) | 32.4 |
| Leucate | 2 | 42.92ºN-3.12ºE | 42.94ºN-3.10ºE | 2.4 (NW) | | 38.2 |
| Barcelona | 3 | 41.32ºN-2.21ºE | 41.31ºN-2.23ºE | 2.1 (SEE) | 0.5 | 76.8 |
| Tarragona | 4 | 41.07ºN-1.19ºE | 41.06ºN-1.19ºE | 0.8 (SW) | 0.5 | 18.2 |
| Dragonera | 5 | 39.56ºN-2.10ºE | 39.56ºN-2.10ºE | 0.5 (NE) | 3 | 183.4 |
| Palma Bay | 6 | 39.49ºN-2.70ºE | 39.48ºN-2.69ºE | 1.9 (SW) | 1 | 31.8 |
| Cala Millor | 7 | 39.59ºN-3.40ºE | 39.60ºN-3.40ºE | 1.5 (NW) | 17 | 17 |
| Son Bou | 8 | 39.90ºN-4.06ºE | 39.90ºN-4.06ºE | 0.5 (SW) | 17 | 17 |
| Málaga | 9 | 36.66ºN-4.44ºW | 36.65ºN-4.44ºW | 1.4 (SSE) | 0.5 | 21.3 |
| Melilla | 10 | 35.32ºN-2.94ºW | 35.35ºN-2.94ºW | 3.4 (NNE) | 0.5 | 16.2 |





*Table 2: Characteristics of the study moorings in the western Mediterranean Sea (name, coordinates of the station and the nearest satellite point, their distance, sensor depth and bathymetry) as displayed in Figure 1. The distance is the one to the nearest satellite point and its orientation from the mooring.*

|  | Maximum Intensity | Cumulative Intensity | Total days |
|---|---|---|---|
| **Sète** | 5-69 | 54-95 | 53-93 |
| **Leucate** | 15-100 | 52-100 | 50-100 |
| **Barcelona** | 17-100 | 64-100 | 65-100 |
| **Tarragona** | 16-100 | 58-100 | 56-100 |
| **Dragonera** | 19-100 | 51-100 | 42-100 |
| **Palma** | 26-100 | 51-100 | 37-100 |
| **CalaMillor** | 20-100 | 55-100 | 43-100 |
| **Son Bou** | 16-100 | 48-100 | 34-100 |
| **Málaga** | 8-100 | 29-100 | 21-100 |
| **Melilla** | 14-100 | 49-100 | 35-100 |

*Table 3. Underestimation error (in %) of annual MHW characteristics (maximum and cumulative intensities, total days) as detected by the nearest satellite points (products ref. no. 1 and 2, Table 1) from moorings (products ref. no. 3, 4 and 5, Table 1) over 2013-2022 with respect to the reference periods 2013-2022 and 1982-2015 (reference for error estimation).*

|  | Mean Intensity | Maximum Intensity | Cumulative Intensity | Duration | Freque ncy | Total days | Total days with T>28ºC [consecutive days] |
|---|---|---|---|---|---|---|---|
| **Sète** | **3.67** <br> 3.28 | 5.11 <br> **5.35** | **146.68** <br> 118.16 | **10** <br> 9 | 4 <br> 4 | **40** <br> 36 | - <br> - |
| **Leucate** | **2.72** <br> 2.49 | **5.17** <br> 3.70 | 212.07 <br> **221.64** | 9.8 <br> **14.8** | **8** <br> 6 | 78 <br> **89** | - <br> - |
| **Barcelona** | 1.80 <br> **2.30** | 2.64 <br> **3.71** | 108.07 <br> **188.23** | 15 <br> **16.4** | 4 <br> **5** | 60 <br> **82** | 8 [6-2] <br> **17** [1-16] |
| **Tarragona** | 2.10 <br> **2.18** | 4.21 <br> 4.22 | **274.48** <br> 242.01 | **21.8** <br> 13.9 | 6 <br> **8** | **131** <br> 111 | **47** [11-19-11-1-4-1] <br> 22 [2-4-15-1] |
| **Dragonera** | 1.87 <br> 1.87 | **3.34** <br> 3.19 | 209.58 <br> **253.11** | 18.7 <br> **27** | **6** <br> 5 | 112 <br> **135** | 53 [1-9-17-26] <br> **56** [7-24-9-10-6] |
| **Palma** | 1.80 <br> **1.91** | 2.45 <br> **2.98** | 221.27 <br> **237.14** | 17.6 <br> **31** | **7** <br> 4 | 123 <br> 124 | 58 [33-25] <br> **59** [43 10 6] |
| **Cala Millor** | 1.85 <br> **1.90** | 3.09 <br> **3.24** | 147.76 <br> **237.71** | **40** <br> 25 | 2 <br> **5** | 80 <br> **125** | 20 [3-4-5-1-6-1] <br> 55 [40-6-3-6] |
| **Son Bou** | 1.90 <br> 1.90 | 2.65 <br> **3.17** | 117.91 <br> **235.27** | **31** <br> 20.7 | 2 <br> **6** | 62 <br> **124** | 8 [5-1-2] <br> 45 [4-29-4-3-3-2] |



| | | | | | | | |
|---|---|---|---|---|---|---|---|
| **Málaga** | **3.51** | 4.38 | **80.69** | 7.7 | 3 | 23 | - |
| | 3.34 | **4.51** | 76.82 | 7.7 | 3 | 23 | - |
| **Melilla** | 1.66 | 2.75 | 77.90 | 9.4 | 5 | 47 | 1 |
| | **1.71** | **3.82** | **168.37** | **12.5** | **8** | **98** | 1 |

*Table 4. Annual MHW characteristics (mean, maximum and cumulative intensities, mean duration, frequency and total days) and number of MHW days with temperature warmer than 28ºC as detected by moorings (products ref. no. 3, 4 and 5, Table 1, in black) and satellite nearest point (product ref. no. 1, Table 1, in red) in 2022.*





**Figures**

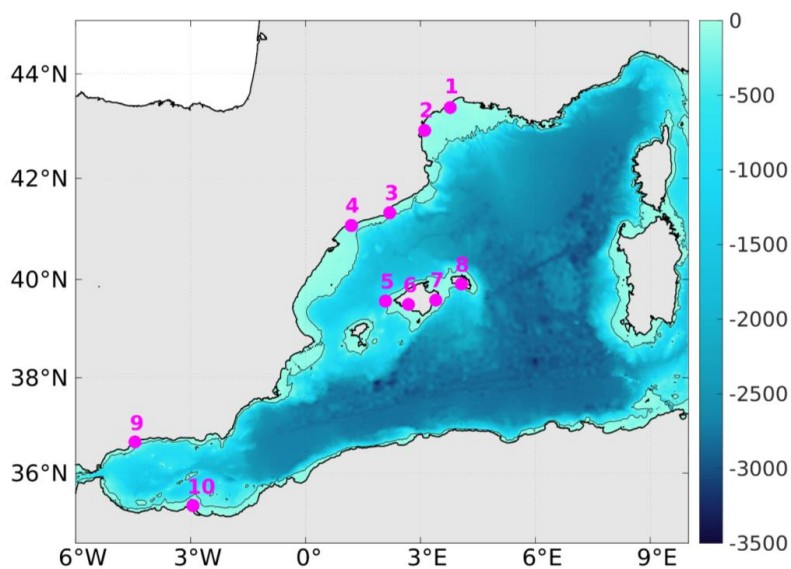


*Figure 1. Bathymetry (in m) in the western Mediterranean Sea with contour at 200m (grey line)*
*and locations of selected mooring for the study (pink points) as listed in Table 2.*

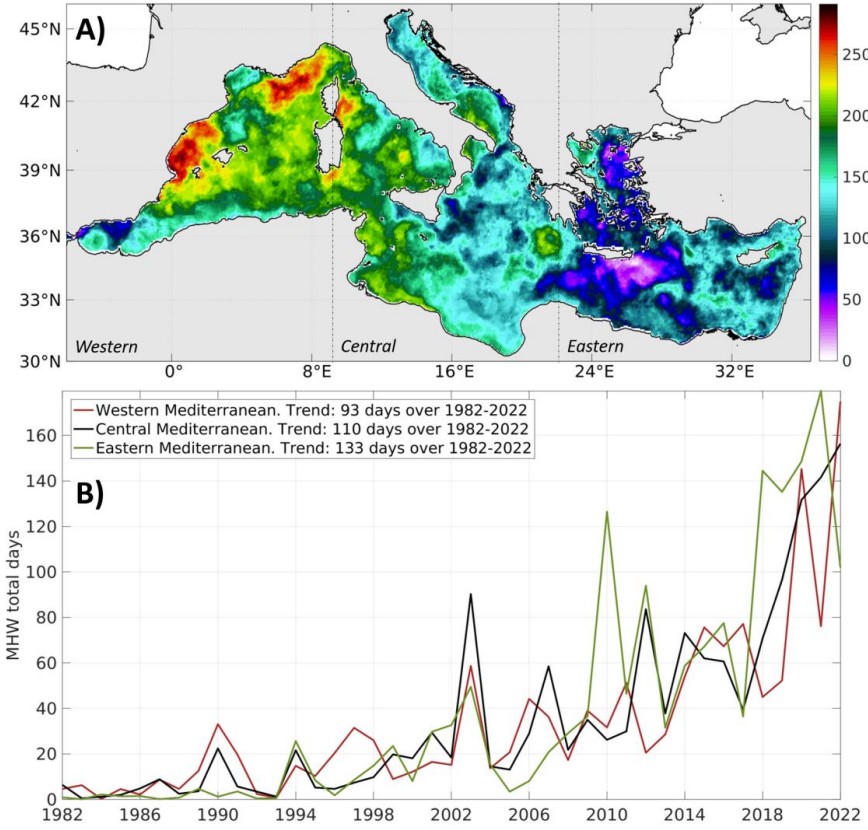


*Figure 2: (A) MHW total days in 2022 from satellite (product ref. no. 1, Table 1) with respect to the historical data (product ref. no. 2, Table 1) over the period 1982-2015. (B) Time series of annual MHW total days averaged in the western, central and eastern Mediterranean sub-basins from 1982 to 2022.*





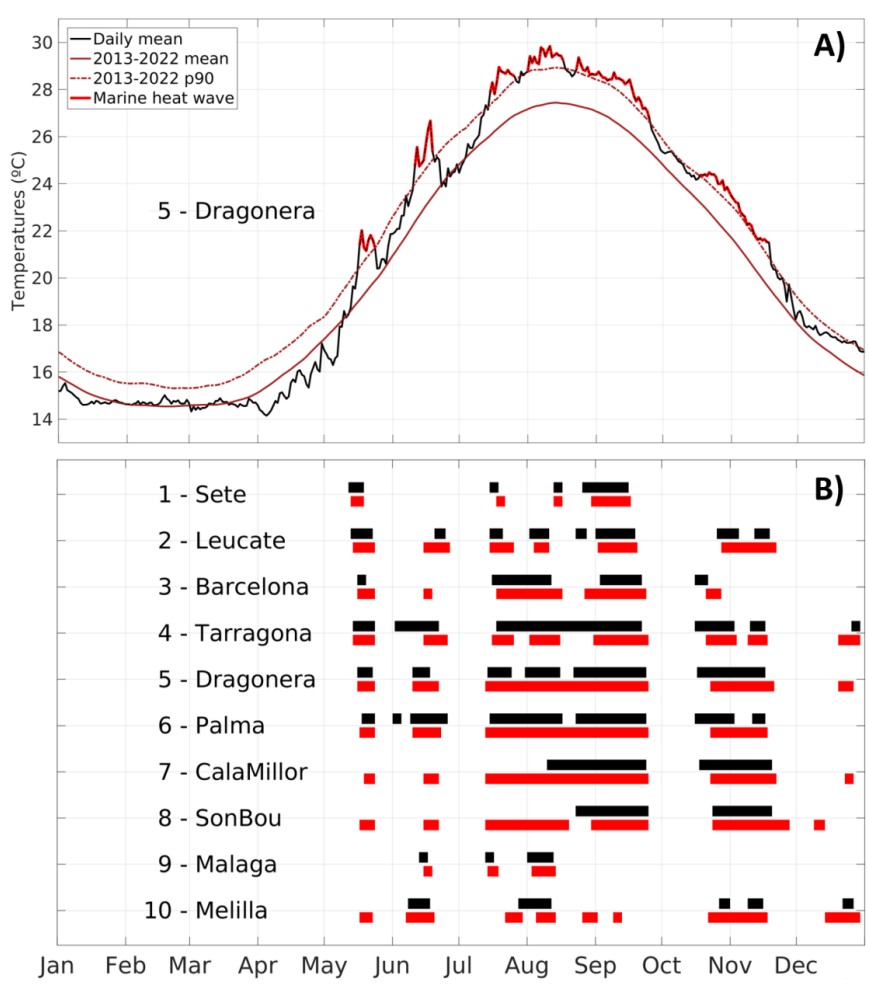

*Figure 3: (A) Daily SST and MHWs from mooring at Dragonera in 2022 with respect to the*
*reference period 2013-2022 (product ref. no. 3, Table 1). (B) MHW days from study moorings*
*(black) and satellites at the nearest point (red) during the year 2022 (products ref. no. 3, 4 and*
*5, Table 1).*



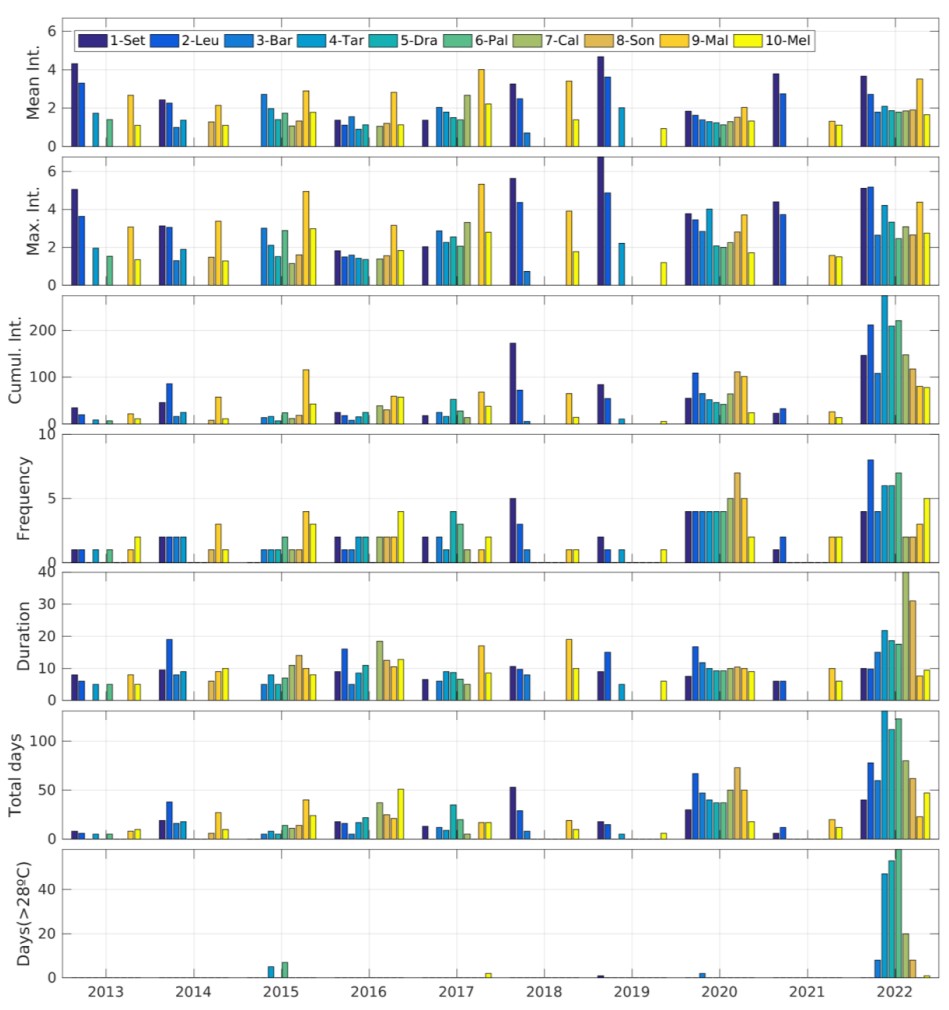

*Figure 4. Annual MHW characteristics (mean, maximum and cumulative intensities, mean*
*duration, frequency and total days) and number of MHW days with temperatures exceeding*
*28ºC as detected by moorings (products ref. no. 3, 4 and 5, Table 1) from 2013 to 2022.*



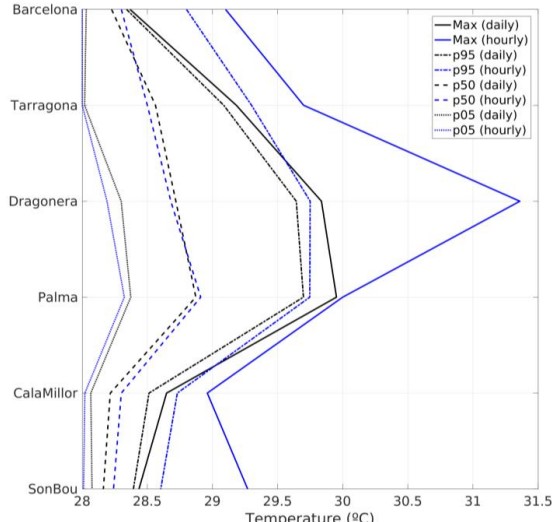


*Figure 5: The 5, 50 and 95th percentiles and maxima of the distribution of MHW temperatures*
*warmer than 28ºC as detected with the daily (black) and hourly (blue) data from moorings*
*(products ref. no. 3, 4 and 5, Table 1).*