# Peer review of "Coastal ocean response during the unprecedented marine heatwaves in the western Mediterranean in 2022"

_State of the Planet, 2023_

## Author Response (AR1)

**Responses to reviewers for the preprint sp-2023-18**

"Coastal ocean response during the unprecedented marine heat waves in the western Mediterranean in 2022" by Juza et al.

**Editor (Piero Lionello)**

The reviewers have read your manuscript and have only minor comments, that you have accepted in your replies in the online discussion. However, I strongly suggest that you improve the abstract and conclusion section to better highlight the value of your study.

>> We thank very much the editor for having revised the manuscript and provided comments for improvements. We considered all of them. Please find the responses below.

Specifically, in the conclusions:

- At Line 254 you write "the ocean response strongly differs from the open ocean to near-shore areas, and from one coastal location to another". This concept is also repeated (avoid repetitions) at line 274 without adding any informative detail.

Please expand:

what are the differences between open ocean and near-shore areas emerging from your study?

Is there a pattern in the differences among locations?

>> We fully agree with the editor and have completed the sentence.

- We provide general statement in the first paragraph of the conclusion. The present study focuses on the coastal areas. A reference has been added to avoid confusion. Indeed, the comparison between the open and coastal water responses was previously performed (using satellite products in near-shore and offshore waters): "The ocean response strongly differs from the open ocean to near-shore areas, and from one coastal location to another (Juza et al., 2022)."

- Then, as required, we provide more detailed information about this study in the coastal waters of the WMed. The statement "Finally, the ocean response to extreme warm events strongly differs from one location to another even in the same region, from surface to subsurface even in very shallow waters." have been modified and complemented as follows: "Finally, the coastal ocean response to extreme warm events strongly differs from north to south WMed. No coincidence is found between north and south nor persistent feature in regional differences.

Coastal MHWs also vary within the same sub-region (Sète-Leucate, Barcelona-Tarragona, Dragonera-Palma, Cala Millor-Son Bou, Malaga-Melilla) where extreme events coincide with differences in intensity and duration both at the surface and in subsurface."

- At line 272 you write "The thermal stress on the physical and biological oceans is also minimized with the use of daily data compared to hourly measurements." Please add a general estimate of the magnitude of such minimization

>> We have complemented the statement as follows: "The thermal stress estimation from high-temperature peaks on the physical and biological oceans is also minimized with the use of daily data which detect underestimated maxima up to 1.52ºC difference during the warm events compared to hourly measurements."

In the abstract (line 14) you generically mention that "Differences between satellite products and mooring observations, as well as between daily and hourly in situ measurements, are highlighted". As such this statement is not very informative and it would be useful to replace or integrate it with a very short description of these differences as they emerge from your study.

>> Following the editor's suggestion, we have modified this statement as follows: "Differences between datasets are also highlighted. In 2022, depending on the sub-regions, satellites underestimated or overestimated MHW duration and intensity compared with *in situ* measurements at the surface. In addition, daily data underestimate maxima reached during the extreme warm events up to 1.52ºC compared with hourly measurements."

Your manuscript is therefore returned for a minor revision (review by editor). Please provide a revised version of your manuscript, that accounts for the comments of reviewer 2, 3 and for my requests, a copy of your manuscript where all your changes are annotated and the list of your detailed answers to the reviewers' comments.

>> We thank the editors for his comments and provided a revised version of the manuscript (where changes appear in color) as well as the detailed responses to editor and reviewers.

**RC1**: Accept in present form

Thank you very much for having revised the manuscript. In the new version, we integrated the following revisions:

- New figure 1: colors of mooring points are aligned with the colors in Figure 4.
- We have also considered the comments of reviewers #2 and #3, adding some clarifications and references.
- A new version of the manuscript is provided with changes indicated.

**RC2**:

Thank you very much for having revised the manuscript and provided comments for improvements. All of them have been considered. Details are provided below. A new version of the manuscript is provided with changes indicated.

In addition, Figure 1 has been improved as follow: colors of mooring points are aligned with the colors in Figure 4.

Line 14: at the surface
>> Thanks. Corrected.

Line 16: the highest values of cumulative intensity and total days were found at the surface
>> Thanks. Corrected.

Line 17: were observed to last up to 58 days
>> Thanks. Corrected.

Line 27: I don't think the references of 2020 and 2021 be can used to explicitly describe the 2022 event. Please make this point clearer.
>> Thank you for rising the lack of clarification. These references are those of the applications (implemented in 2020 and 2021) that are operational and provide continuously updated information. To be clearer, I have modified and completed the sentence as follow: "as displayed in operational applications (Juza and Tintoré, 2020, 2021b) and recently reported (Marullo et al, 2023)"

Line 33: be found more locally than regionally
>> Thanks. Corrected.

Line 153: Doesn't make sense.

>> Completely right, thanks, corrected: "In 2022, all moorings of the coastal WMed detected MHWs over the last decade (Figure 3), although MHWs were computed using the reference period 2013-2022.

Line 188: moorings
>> Thanks. Corrected.

Line 190: I agree with the statements in this paragraph but potential bias between the datasets should also be considered.
>> We have completed the statement considering the suggestion of the reviewer: "Differences between MHWs detected by satellites and moorings may be explained by several factors such as the sensor or platform type, spatial and temporal coverage, specific bias at a particular platform, instrumental corrections, validation and calibration, interpolation methods as well as the effective depth of measurements (Alvera-Azcárate et al., 2011). "

Alvera-Azcárate, A., Troupin, C., Barth, A., & Beckers, J. M. (2011). Comparison between satellite and in situ sea surface temperature data in the Western Mediterranean Sea. Ocean Dynamics, 61, 767-778, doi:10.1007/s10236-011-0403-x

Line 198: highlighting how the coastal ocean response differs
>> Thanks. Corrected.

Line 213. While only one season was concerned in 2013
>> Thanks. Corrected.

Line 213: which seasons?
>> Completed: "While one season was concerned in 2013 (summer or autumn depending on the mooring), MHW occurrences covered three seasons in 2022 (mainly spring, summer and autumn) (not shown)"

Line 275: You mean the thermal stress presented by T>28? This should be made clear.
>> Right, thermal stress generated by extreme warm temperatures and peaks. I would like to provide a more general sentence in conclusion, I suggest the following sentence to be clearer: "The thermal stress estimation from high-temperature peaks on the

physical and biological oceans is also minimized with the use of daily data compared to hourly measurements."

**RC3**: I agree with the comments posted by Referee #2 in the interactive discussion section and I will not rewrite them here. In my opinion, paper is between "Accept in present form" and "very minor revisions". As final comment I suggest to the authors to add two more reference: Pisano et al. at the beginning of the introduction and Marullo et a. 2023 in the methodology section where is written "…..recent publications in the Mediterranean Sea (Juza and Tintoré, 2021; Juza et al., 2022,…..).

Pisano, A., Marullo, S., Artale, V., Falcini, F., Yang, C., Leonelli, F. E., ... & Buongiorno Nardelli, B. (2020). New evidence of Mediterranean climate change and variability from sea surface temperature observations. Remote Sensing, 12(1), 132.

Marullo, S., Serva, F., Iacono, R., Napolitano, E., di Sarra, A., Meloni, D., ... & Santoleri, R. (2023). Record-breaking persistence of the 2022/23 marine heatwave in the Mediterranean Sea. Environmental research Letters.

Thank you very much for having revised the manuscript and suggesting two references that have been added as follow:
- Pisano et al., 2020 have been added at the beginning of the introduction, as suggested by the reviewer
- Marullo et al, 2023 have been added in the introduction when stating about the exceptional MHWs in 2022 rather than in the methodology (where we were referring to the baseline period which is different from Marullo et al., 2023)

We also congratulate the authors for their new publication.

We have also considered all the comments of Reviewer#2. All changes are indicated in the new version of the manuscript.

Finally, we have also improved Figure 1 as follow: colors of mooring points are aligned with the colors in Figure 4.